# N-aryl pyrido cyanine derivatives are nuclear and organelle DNA markers for two-photon and super-resolution imaging

Kakishi Uno [1,3,4], Nagisa Sugimoto [2] & Yoshikatsu Sato [1,2,4✉]

Live cell imaging using fluorescent DNA markers are an indispensable molecular tool in various biological and biomedical fields. It is a challenge to develop DNA probes that avoid UV light photo-excitation, have high specificity for DNA, are cell-permeable and are compatible with cutting-edge imaging techniques such as super-resolution microscopy. Herein, we present N-aryl pyrido cyanine (N-aryl-PC) derivatives as a class of long absorption DNA markers with absorption in the wide range of visible light. The high DNA specificity and membrane permeability allow the staining of both organelle DNA as well as nuclear DNA, in various cell types, including plant tissues, without the need for washing post-staining. N-aryl-PC dyes are also highly compatible with a separation of photon by lifetime tuning method in stimulated emission depletion microscopy (SPLIT-STED) for super-resolution imaging as well as two-photon microscopy for deep tissue imaging, making it a powerful tool in the life sciences.

---

[1] Graduate School of Science, Nagoya University, Nagoya, Japan. [2] Institute of Transformative Bio-Molecules (WPI-ITbM), Nagoya University, Furo, Chikusa, Nagoya, Japan. [3] Present address: Department of NanoBiophotonics, Max Planck Institute for Biophysical Chemistry, Göttingen, Germany. [4] These authors contributed equally: Kakishi Uno, Yoshikatsu Sato. ✉email: sato.yoshikatsu@i.mbox.nagoya-u.ac.jp

Synthetic fluorescent dyes that stain DNA are widely used in life sciences research. In addition to gel electrophoresis[1], polymerase chain reaction (PCR) in molecular biology[2], and flow cytometry[3], they are also greatly used in cell biological study such as for visualizing nuclear and organelle DNA[4,5], cell proliferation analysis[6], and diagnosis of viral infection[7]. The ideal properties for a synthetic DNA marker in living cells include (i) high specificity of DNA over ribonucleic acid (RNA), (ii) the applicability to long wavelength photoexcitation (>532 nm)[8], and (iii) the ability to stain DNA in diverse living cells and tissues. A marker with these properties, however, has not yet developed.

The blue emitting Hoechst 33342[9] is currently the most popular DNA marker although it requires phototoxic UV light for photoexcitation[10]. To overcome this drawback, the Hoechst tagging strategy, fusing a fluorescent dye excited by visible wavelength to Hoechst through a linker, has been recently reported[11,12]. Especially, silicon-rhodamine (SiR)-Hoechst, which fused a far-red emissive SiR to Hoechst 33342, stained the nucleus with no significant phototoxicity and is compatible with stimulated emission depletion (STED) nanoscopy[12–14]. In exchange for these preferable properties, however, the linking of the additional fluorescent dye decreases the DNA-binding strength of Hoechst[12,14].

Unsymmetrical cyanine fluorescent dyes, such as SYBR-Green I (SG)[15] and Pico-Green (PG)[16], are also widely used as DNA markers. These dyes, unlike Hoechst, can also stain mitochondrial DNA (mt-DNA) in addition to nuclear DNA in living cells[5,17–19]. Unfortunately, however, these dyes are mostly excited with phototoxic short wavelength lasers such as the 488 nm laser[8] and do not have a high specificity for DNA over RNA[20,21]. So far, a fluorescent DNA marker which fulfills all the desirable properties is not available.

In the present study, we introduce a series of long wavelength (500 to 700 nm) absorptive DNA-binding dyes, containing a pyrido cyanine (PC) backbone. In addition to not requiring UV light excitation, they also display great cell permeability in various cell types and exceptional high DNA specificity for clear DNA

staining of nuclear and organellar DNA in living cells. Furthermore, we demonstrate their compatibility with various optical microscopes such as two-photon excitation microscopy (2PEM), fluorescent lifetime imaging microscopy (FLIM), and separation of photon by lifetime tuning (SPLIT) method in STED nanoscopy (SPLIT-STED)[22].

## Results

**Molecular design and in vitro characterization of N-aryl PC derivatives.** To achieve long absorption DNA selective markers with a small-molecule fluorophore, we envisioned that PC core could serve as a useful backbone due to its extended π-conjugation and the extendable methine unit[23,24]. As the result of several investigation, we discovered that the N-aryl moiety is an essential component for DNA binding. We first synthesized **PC1** and examined its photophysical properties (Fig. 1a). UV-Vis and fluorescence spectra measurements revealed that the maximum absorption wavelength of free **PC1** was 532 nm upon binding to DNA (Table 1 and Supplementary Fig. 1a). The fluorogenic properties ($I^{dsDNA}/I^{free}$) of **PC1** increased 1600-fold upon binding to DNA, whereas only 110-fold increase was observed upon binding to RNA ($I^{RNA}/I^{free}$) (Table 1 and Supplementary Fig. 2a). The value of DNA/RNA specificity ($I^{dsDNA}/I^{RNA}$) of **PC1** was higher than both Hoechst 33342[25] and PG[21] (Supplementary Fig. 2b, j, k). Fluorescence titration with three hairpin oligonucleotides ($^{AATT}DNA$, $^{CCGG}DNA$, and $^{AAUU}RNA$)[12,26] revealed that **PC1** binds preferentially to $^{AATT}DNA$, while binding to $^{CCGG}DNA$ and $^{AAUU}RNA$ was not observed (Fig. 1c and Supplementary Fig. 3). These results suggest that **PC1** specifically bind to the AT base-pair of nucleic acids. We also examined competitive experiment in DNA sequence using $^{AATT}DNA$ hairpin oligo and found that **PC1** shares the same $^{AATT}DNA$ sequences with Hoechst (Supplementary Fig. 3k). Furthermore, to assess the binding mode of **PC1** to double stranded DNA (dsDNA), we measured circular dichroism (CD) spectrum of **PC1** complexed with dsDNA. The CD spectrum clearly showed a positive cotton effect appeared at the

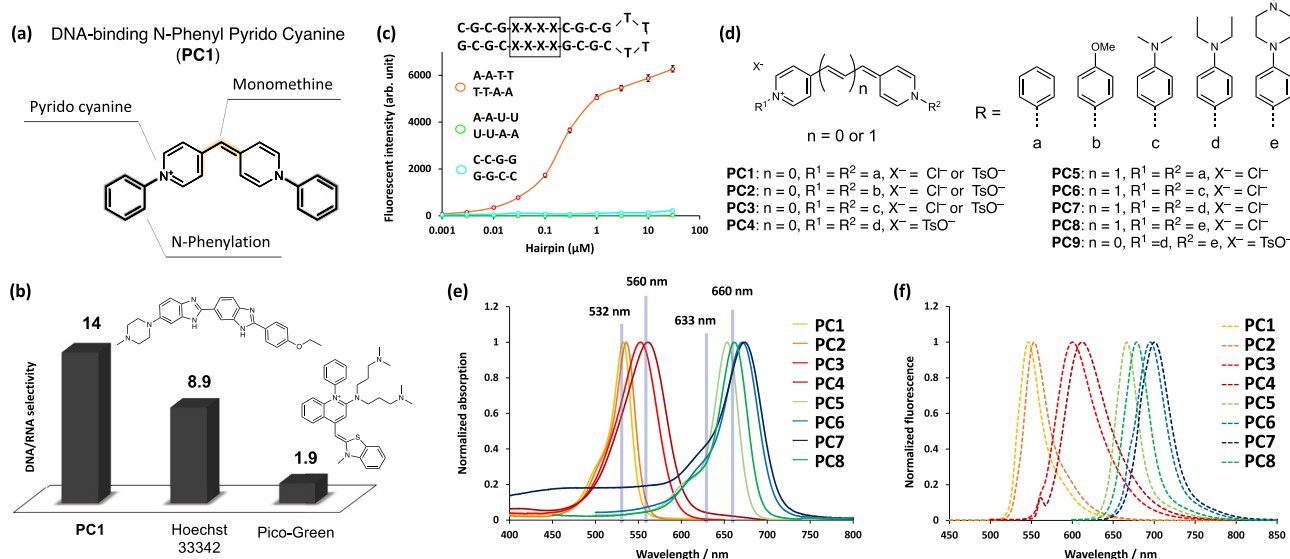

**Fig. 1 Molecular design and in vitro characterization of N-aryl PC derivatives. a** The structure and structural components of **PC1**. **b** The comparison of DNA/RNA selectivity of **PC1**, Hoechst 33342, and PG. **c** The titration curve of 100 nM **PC1** with various concentration of hairpin oligonucleotides. Fluorescence intensity is expressed in arbitrary units (arb. u.). The error bars indicate means ± s.d. of three independent replicates. Source Data is available as a source data file. **d** The general structure of PC dyes and their substituent patters with corresponding compound names; the unit of methylene length and N-aryl groups are represented as "n" and "R" respectively. The normalized absorption (**e**) and fluorescence spectra (**f**) of all PC dyes when complexed with calf thymus double stranded DNA (dsDNA) in tris-EDTA buffer solution. (pH = 8.0); see details in Figs. S1, S2.

**Table 1 Photophysical properties of all synthesized N-aryl PC derivatives.**

| Name | $\lambda_{abs}^{dsDNA}$/nm (a) ($\lambda_{abs}^{free}$/nm) (b) | $\varepsilon^{dsDNA}$/$10^4$ $M^{-1}cm^{-1}$ (c) | $\lambda_{em}^{dsDNA}$/nm (d) ($\lambda_{em}^{free}$/nm) (e) | $\Phi_F^{dsDNA}$(f) ($\Phi_F^{free}$) (g) | $I^{dsDNA}/I^{free}$ (h) | $I^{RNA}/I^{free}$ (i) | $I^{dsDNA}/I^{RNA}$ (j) | $\tau^{dsDNA}$/ns (k) |
|------|------|------|------|------|------|------|------|------|
| PC1 | 532 (510) | 14 | 546 (535) | 0.09 (0.0004) | 1600 | 110 | 14 | 0.62 |
| PC2 | 536 (512) | 13 | 553 (537) | 0.22 (0.0006) | 2500 | 170 | 15 | 0.86 |
| PC3 | 552 (520) | 8.6 | 600 (592) | 0.42 (0.0018) | 3900 | 130 | 31 | 1.5 |
| PC4 | 561 (524) | 7.7 | 612 (605) | 0.44 (0.0026) | 1700 | 53 | 32 | n.d. |
| PC5 | 654 (630) | 15 | 666 (652) | 0.4 (0.039) | 140 | 25 | 5.7 | 2.7 |
| PC6 | 671 (639) | 10 | 695 (666) | 0.26 (0.038) | 54 | 2.1 | 26 | 1.7 |
| PC7 | 674 (642) | 9.6 | 700 (683) | 0.20 (0.038) | 57 | 2.3 | 25 | 1.8 |
| PC8 | 662 (637) | 9.7 | 678 (666) | 0.33 (0.075) | 39 | 20 | 2 | 2.3 |

Tris-EDTA buffer solution (TE buffer, pH 8.0) was used for all optical measurements.
[a]Maximum absorption wavelength of dye-dsDNA complexes.
[b]Maximum absorption wavelength of dyes in their free states.
[c]Molar absorption coefficient of dye-dsDNA complexes.
[d]Maximum fluorescence wavelength of dye-dsDNA complexes.
[e]Maximum fluorescence wavelength of free dyes.
[f]Fluorescence quantum yields of dye-dsDNA complexes.
[g]Fluorescence quantum yield in their free states.
[h, i]Ratio of fluorescence increases of each dye upon binding to dsDNA and RNA; see the SI for the detailed changes in fluorescence spectra.
[j]The value of calculated DNA/RNA selectivity (see main text).
[k]Fluorescence lifetime of dye-dsDNA complexes.

maximum absorption band, similar to that of DNA minor-groove binders, Hoechst[27] and DAPI[28] (Supplementary Fig. 4a). This result suggests that **PC1** binds to dsDNA with minor-groove. The similar binding mode of Hoechst would contribute the high selectivity of **PC1** to DNA over RNA. Next, to investigate the higher DNA selectivity of PC dyes (**PC1**–**4**) than Hoechst's, we synthesized a PC dye with a (methyl)piperazine site by replacing one of the two diethylamino groups of **PC4** with (methyl)piperazine and examined the fluorogenic properties (**PC9**, Fig. 1d and Supplementary Fig. 2i). As a result, the fluorescence intensity of **PC9** increased 250-fold upon binding RNA, which was five times higher than that of **PC4** (53-fold) (Supplementary Fig. 2d, i). This result suggests that the binding of the (methyl) piperazine moiety to RNA is thought to be a factor that reduces DNA selectivity.

To achieve further optical red-shift, we introduced various N-aryl groups[29] as well as extended monomethine to trimethine[30] (Fig. 1d). By selecting the proper N-aryl groups and methine lengths, the absorption wavelength of N-aryl PC dyes can be shifted from 500 to 700 nm, providing researchers with many color options (Fig. 1e, f). Importantly, the high DNA/RNA selectivity and high DNA-based fluorogenic properties are not impaired by substitution with various N-aryl groups (Table 1 and Supplementary Figs. 1, 2). We revealed that **PC5** specifically binds to AT-pairs (Supplementary Fig. 3f) and the CD spectrum showed a strong positive cotton effect (Supplementary Fig. 4b). This result suggests that **PC5** binds to dsDNA with minor-groove, playing a part in having comparable DNA/RNA specificity, unlike SG backbone[30].

**Application of N-aryl PC dyes in various living-cell types.** To examine N-aryl PC dyes for nuclear marker, we stained HeLa cells with each synthesized N-aryl PC dyes at a concentration of 1 µM and found that they were able to label nuclear DNA (Supplementary Fig. 5). To determine the binding specificity of N-aryl PC dyes in live cell nucleus, we co-stained the AT-rich sequence binding dye, Hoechst 33342[25,26] with **PC1** or **PC3** (Supplementary Fig. 6). We found that the N-aryl PC dyes were excluded from the nucleus by Hoechst 33342 in a dose dependent manner. These results indicate that they scramble for AT sequences pairs with Hoechst, which is consistent with our in vitro study using hairpin oligonucleotides (Fig. 1c and Supplementary Fig. 3). The ability of N-aryl PC dyes to selectively stain DNA without a washing process was compared with commercially available cell

permeable SYTO dyes, which have similar excitation and emission spectra. **PC1** and **PC3** clearly stain the nucleus and chromosomes with little background from cytosol, whereas the SYTO dyes failed to stain the nucleus even at higher concentrations. The SYTO dyes instead stained the cytosol and nucleolus, which contain large amounts of RNA (Fig. 2a and Supplementary Fig. 7). Moreover, cell proliferation analysis and corresponding time-lapse imaging revealed that **PC1** and **PC3** showed little cytotoxicity and phototoxicity compared with SYTO dyes (Fig. 2b). **PC1** and **PC3** were also investigated using other mammalian cell types (U-2OS, C6, and NIH3T3). We found that **PC1** and **PC3** could uniformly stain nuclear DNA without addition of a reagent to facilitate cell permeability (Fig. 2c). Considering that SiR-Hoechst requires the voltage-dependent calcium channel inhibitor, verapamil, for homogeneous staining of nuclear DNA in U-2OS cells[12,14], these results indicate that cell permeability of N-aryl PC dyes is substantially higher than that of SiR-Hoechst. Next, we investigated the ability of N-aryl PC dyes to stain the nucleus of plant tissues which are composed of multicell layers with thick cell walls. We demonstrated this using the Arabidopsis leaf and root tissues (Fig. 2d). The epidermal cells including the stomata and mesophyll cells underneath the epidermis were completely stained in the nucleus of the leaf tissue in the absence of any washing steps (Supplementary Movies 1 and 2). Time-lapse analysis revealed that the root hairs as well as the main root tissue grew normally and were also clearly stained in their nucleus by **PC1** (Supplementary Movie 3). We next employed 2PEM excited with 1000 nm to investigate the dye penetration in the main root. Using 2PEM, we observed the entire nuclei of the root tip region by overall cross optical sections (~100 µm) while single photon excitation with 488 nm enabled visualization of only the half cross sections (~50 µm) (Fig. 2e and Supplementary Movie 4). We did not detect any cytotoxicity or phototoxicity in the plant cells stained by PC dyes. Cell division with root growth was frequently observed using 2PEM time-lapse imaging (Fig. 2f and Supplementary Movie 5). These results indicate that the N-aryl PC dyes penetrated deep into the root layer with no apparent toxicity and are very suitable for 2PEM possibly because of the symmetrical donor–acceptor–donor pi-conjugation of PC dyes[31].

**Discriminating between nuclear DNA and organellar DNA using fluorescence lifetime of N-aryl PC dyes.** Even at only 10

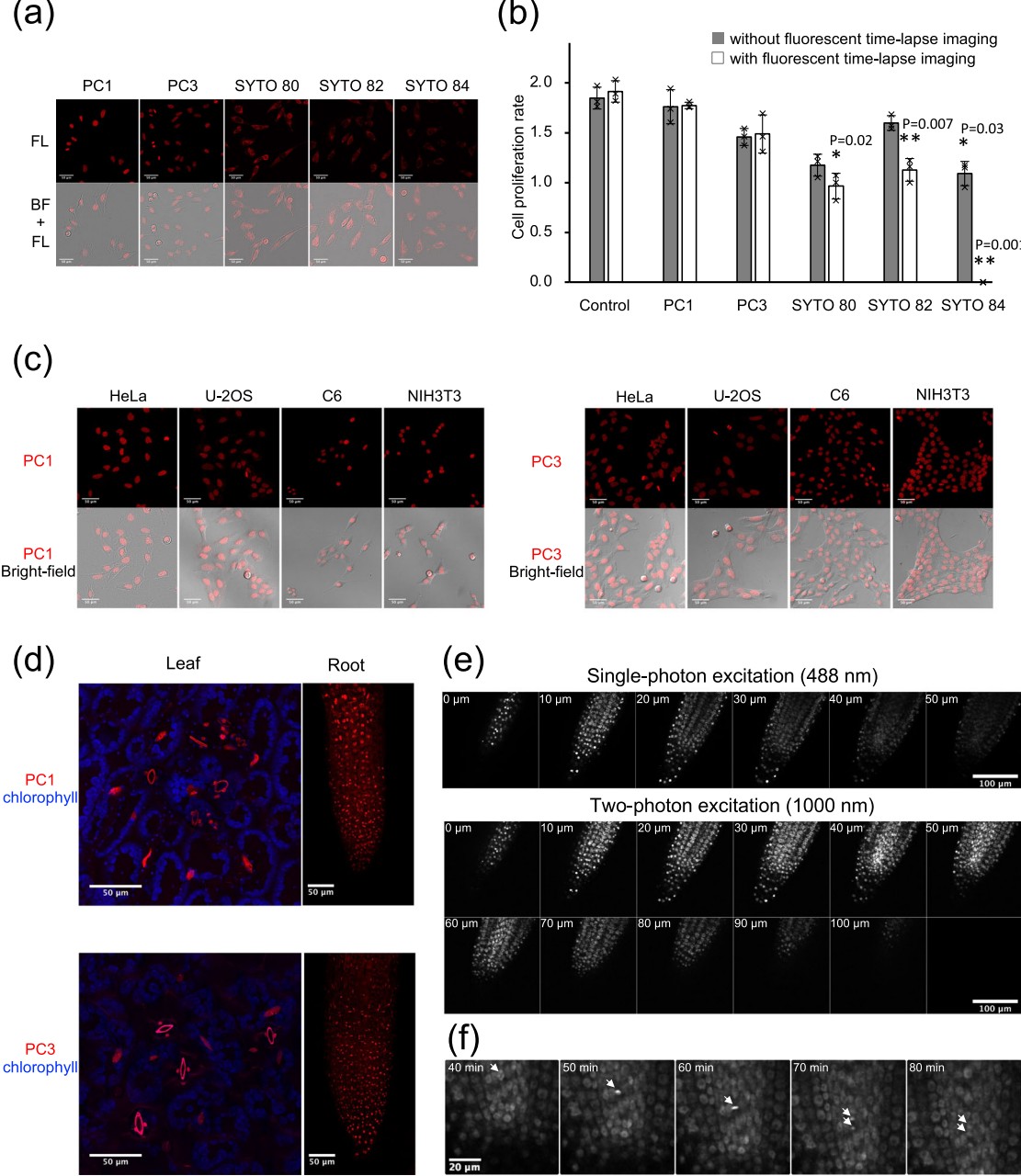

**Fig. 2 Live cell imaging with PC1 and PC3. a** Live cell fluorescent microscopy with **PC1**, **PC3**, and commercialized red fluorescent DNA dyes. HeLa cells were stained with each dye at 100 nM. Images are maximum z-projections of total planes (1 μm intervals). **b** Quantification of cell proliferation rate of cells stained with DNA labeling dyes. HeLa cells were stained with each dye at 30 nM and observed every 5 min with z-sectioning (six frames at 3 μm steps) for 24 h. The proliferation rate was quantified as fold changes based on the number of cells between the first frame (0 h) and the last frame (24 h) of bright-field images. Gray bars indicate the results from only bright-field time-lapse imaging without fluorescent time-lapse imaging and white bars indicate the results from both bright-field and fluorescent time-lapse imaging. Error bar shows mean ± s.d. from three independent biological replicates (>25 cells per replicate). Individual data points were also indicated in black cross. Statistical significance (*$P < 0.05$, **$P < 0.01$) of difference from control condition was examined by two-sided student $t$-test using Microsoft Excel. Source Data is available as a source data file. **c** Live cell fluorescent images of different culture cell types with 500 nM **PC1** and **PC3**. The images are maximum z-projections of total planes (1 μm intervals) The representative images are shown from at two to three similar images. **d** Live cell fluorescent images of Arabidopsis leaf and root cells with 1 μM **PC1** and **PC3**. The images are maximum z-projections of total planes (1.1 μm intervals) The representative images are shown from five to six similar images. **e** Comparison of imaging penetration for single and two-photon excitation microscopy in Arabidopsis root tip stained with 1 μM **PC1**. Images were shown every 10 μm steps from z-sectioning images at 1 μm interval. The representative images are shown from four similar images. **f** Time-lapse observation by two-photon microscopy excited with 1000 nm in Arabidopsis root stained with 5 μM **PC1**. Root tip was observed every 2 min with z-sectioning (50 frames at 2 μm steps). The representative images are shown from four similar images.

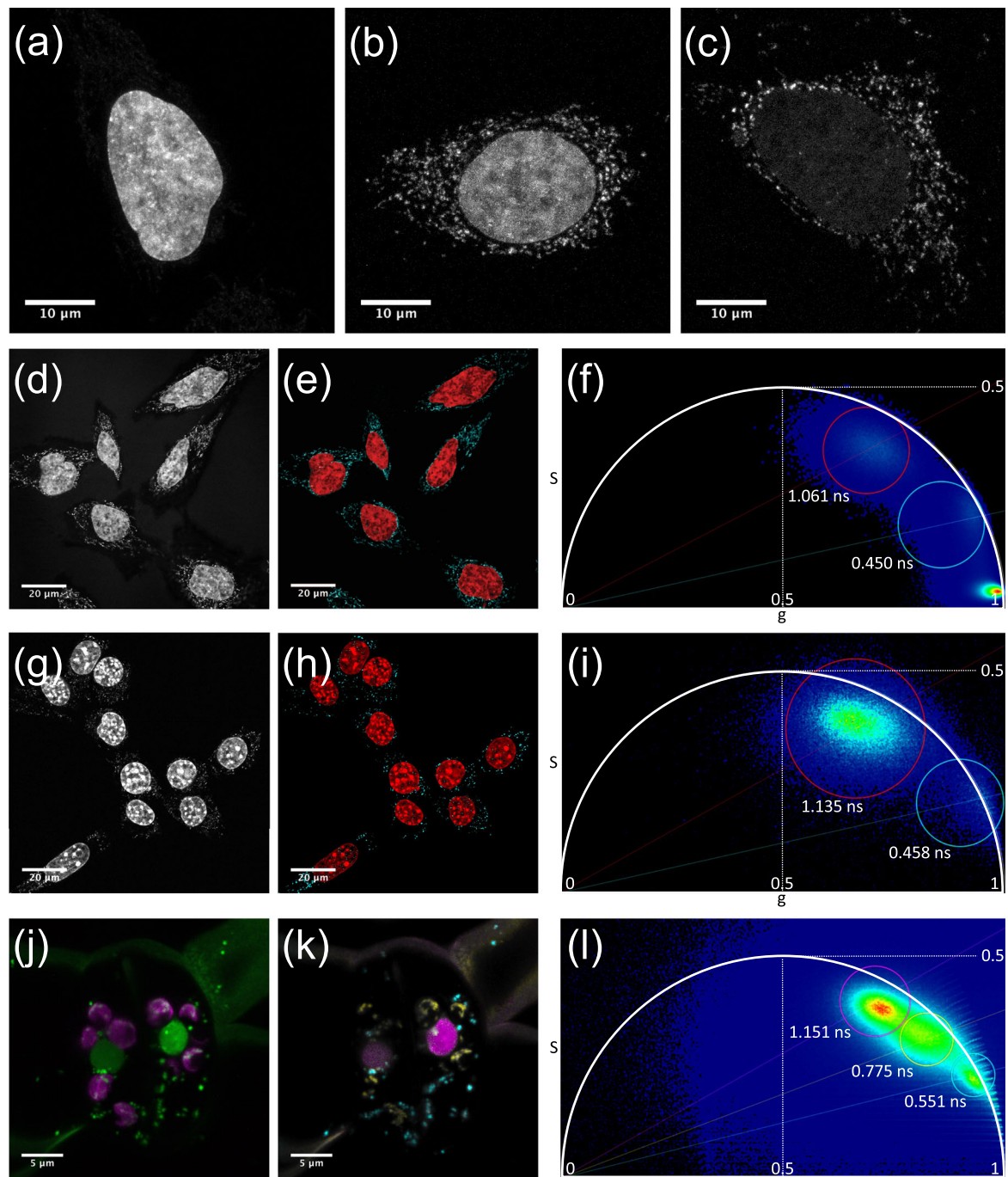

**Fig. 3 Discrimination between nuclear DNA and mt-DNA with fluorescence lifetime of PC1. a–c** Concentration dependence of staining pattern with **PC1**. **a** 10 nM, **b** 1 nM, **c** 100 pM. The images are maximum z-projections of total planes (0.3 μm intervals). The representative images are shown from three to four similar images in each concentration. **d–l** Fluorescent intensity images (**d**, **g**, **j**) and FLIM based separation images of nuclear DNA and mitochondrial DNA (**e**, **h**, **k**) by phasor plot analysis (**f**, **i**, **l**). The pseudo colors of (**e**, **h**, **k**) correspond to the colors of circles in (**f**, **i**, **l**). The nuclear DNA, mt-DNA, and ch-DNA are shown in red, cyan, and yellow, respectively. HeLa cells (**d–f**) and NIH3T3 (**g–i**) were stained with 1 nM and 10 nM **PC1**, respectively and the fluorescent spectrum were collected between 540–650 nm excited at 532 nm. The representative images are shown from 5 and 13 similar images, respectively. Stomata in Arabidopsis leaf cells was stained with 300 nM **PC1** and excited at 532 nm. The fluorescent spectrum of **PC1** and chlorophyll autofluorescence were collected between 540–620 nm and 680–700 nm shown in green and magenta in (**k**), respectively. The representative images are shown from similar images of ten guard cells.

nM, **PC1** stained the nucleus with very little cytoplasmic background (Fig. 3a). In contrast, many cytoplasmic spots were also observed at a concentration of 1 nM and these spot signals became predominant by further dilution of **PC1** to 100 pM (Fig. 3b, c). These results are similar to our previous results using

SG, which stained not only nucleus but mitochondrial nucleoids (mt-nucleoids), the core complexes of mt-DNA replication and transcription[19]. Next, we confirmed that the fluorescence spots resided in mitochondria by co-staining with **PC1** and Mito-Tracker dye and nicely overlapped with those of SG

(Supplementary Fig. 8). These results indicate that **PC1** enables to stain mt-DNA as well as nuclear DNA in a dose dependent manner. Since the synthetic probes for mt-DNA have so far been limited to DAPI[32], SG[5,17], and PG[18], red fluorescent N-aryl PC dyes will provide researchers with an additional color channel for mt-DNA imaging. Furthermore, we found that the fluorescent lifetime of **PC1** in the nucleus (~1.1 ns) was substantially longer than that observed in mt-nucleoids (~0.5 ns). The shorter fluorescent lifetime in mt-nucleoids can be due to the self-absorption because PC dyes with short stokes-shifts are cationic and tend to accumulate in mitochondria (Supplementary Fig. 9)[33,34]. Taking advantage of these properties, we employed FLIM and phasor plot analysis to discriminate between nuclear DNA and mt-DNA in various cell types (Fig. 3d–l). Similar results were obtained in the **PC3** stained cells (Supplementary Fig. 10). Then, we envisioned that the FLIM analysis combined with N-aryl PC dye staining could in plant cells also distinguish chloroplast DNA (ch-DNA), from mt-DNA and nuclear DNA. Next, we employed FLIM analysis of **PC1** in living stomata of Arabidopsis and found that **PC1** in chloroplasts (yellow) had an intermediate fluorescence lifetime (~0.9 ns) compared with that in mitochondria (cyan) and in nucleus (red) (Fig. 3k and Supplementary Movie 6). Thus, FLIM analysis stood out the talented ability of PC dyes, where we successfully demonstrated that **PC1**, on its own achieved to discern all three DNA containing organelles (nucleus, mitochondria, and plastid) in eukaryotes.

**Application of N-aryl PC dyes for live-cell SPLIT-STED nanoscopy.** To our knowledge, probes detecting mt-nucleoids that are compatible with live cell super-resolution microscopy have not been reported to date, even though there is the great need for probes to examine mt-nucleoids with sizes smaller than the diffraction limit of visible light[35,36]. Therefore, we evaluated the compatibility of N-aryl-PC dyes with SPLIT-STED nanoscopy. We firstly confirmed that live-cell SPLIT-STED nanooscopy in HeLa cells stained with 1 μM **PC3** showed detailed chromatin structures with smaller than a diffraction limit. (Supplementary Fig. 11). Then, we applied SPLIT-STED nanoscopy to mt-nucleoids in HeLa cells stained with 10 nM **PC3** and achieved to visualize mt-nucleoids at a much smaller size than the visualization in conventional confocal microscopy (Fig. 4a, b). Similar results were also obtained with live NIH/3T3 cells as well as with plant cells (Supplementary Figs. 12, 13). Next, we calculated a full width at half maximum (FWHM) of minor axis of mt-nucleoids to estimate the size of mt-nucleoids in HeLa cells (Fig. 4c). The FWHM value was 100 ± 9 nm at 3 ns of delay time, which is good agreement with the previous study (Fig. 4d)[37,38]. Taken together, these results indicate that N-aryl PC dyes have a great compatibility with STED nanoscopy in various cell types and useful tool for the study of nuclear and mt-DNA in living cells.

## Discussion

There has been a growing appreciation of the role of DNA probes without UV excitation for live imaging. In this study, we developed a series of DNA markers with a PC backbone to fulfill all three required properties described above. Firstly, we actualized that all N-aryl PC dyes were applicable to long wavelength excitation over 532 nm as planned. This achievement is due to our molecular design focusing on PC[23], possessing an extended π-conjugation with a monomethine cyanine unit compared with hitherto unsymmetrical cyanine dyes such as SG and PG. Secondly, all eight N-aryl PC dyes including trimethine dyes have high DNA selectivity and are capable of staining nucleus. Among them, **PC1** and **PC3** have an unexpectedly great property of DNA sequence selectivity at AT-pairs rich region and a high

fluorogenic property upon binding DNA, succeeding in staining nuclear DNA at very low concentration. And thirdly, these dyes exhibited high cell and tissue permeability to stain nuclear DNA in various cell types. Especially in **PC1** stained plant root, 2PEM revealed whole nuclei without apparent toxicity in root growth and cell division.

In addition to the above basic properties, we also succeeded to bring out other key properties of N-aryl PC dyes. We found that N-aryl PC dyes obviously stained mt-DNA at ultralow concentration and also labeled both mt-DNA and nuclear DNA by optimizing the staining dye concentration. Since the synthetic probes for mt-DNA have been limited to staining DAPI[32], SG[5,17], and PG[18], our red fluorescent PC dyes will provide researchers with an additional optical channel. More importantly, by using FLIM, we also successfully revealed that N-aryl PC dyes, on their own, allow to discriminate between mt-DNA and nuclear DNA in mammalian cells and also to distinguish all three types in plant cells. Furthermore, we demonstrated that N-aryl PC dyes were applicable to SPLIT-STED nanoscopy. Therefore, our dye will be useful for the biology field in mt-nucleoid as well as nuclear chromatin dynamics. The recent work investigating the structural relationship between mt-nucleoids and mitochondrial cristae revealed that super-resolution imaging of cristae was perfectly performed using SNAP-Cell Sir labeling system but the use of PG did not allow the revealing of structural detail of mt-nucleoids beyond the optical resolution[39]. We hope that N-aryl PC dyes would help to achieve dual STED nanoscopy in the challenging study. Last of all, together with these useful characters and talented applicability to various microscope techniques, we believe that N-aryl PC dyes would serve as powerful molecular tools in molecular and cellular biology research.

## Methods

**Binging of hairpin DNA and RNA oligonucleotides.** For the oligonucleotides binding studies, synthetic DNA oligonucleotides, 5′-CGCGAATTCGCGTTTTCG CGAATTCGCG-3′ (28 bp) and 5′-CGCGCCGGCGCGTTTTCGCGCCGGCGC G-3′ (28 bp) were purchased by eurofins, whereas RNA 5′-CGCGAAUUCGCG UUUUCGCGAAUUCGCG-3′ (28 bp) was obtained from FASMAC. Each oligonucleotide was dissolved in 1× TBS buffer (50 mM Tris HCl, 150 mM NaCl, pH 7.4) at 200 μM concentration and adjusted various concentrations by serial dilution with TBS. These oligonucleotide solutions were heated at 95 °C for 1 min followed by cooling down at room temperature. On the other hand, each PC dye was dissolved in TBS with 2 mg/mL BSA at 200 nM concentration. The fluorescent intensity of equal amount mixture of oligonucleotide and PC dye solution was measured by EnSpire (PerkinElmer). Each dye was excited at its absorption peak wavelength and fluorescence was detected at the fluorescent peak wavelength.

**Fluorescent titration.** Calf thymus dsDNA, purchased from Sigma-Aldrich Co, and RNA from torula yeast, purchased from Wako Pure Chemical Industries, were used in the fluorescence titration[40,41]. DNA/RNA selectivity of **PC1** was compared with commercialized nucleus markers (Pico-Green, Hoechst 33342; ThermoFisher Scientific). All chemicals are used without additional treatment or further purifications. UV/Vis absorption spectra were recorded on a Shimadzu UV-3510 spectrometer with a resolution of 0.5 nm and emission spectra were measured with an FP-6600 Hitachi spectrometer with a resolution of 0.2 nm. CD spectra were measured with a JASCO FT/IR6100. 1.0 cm square quartz cell was used for all optical measurements. About 1.0 g/L dsDNA solution (1.0 mL) or 2.0 g/L RNA solution (1.0 mL) were added to dye solutions (2.0 mL) with absorbance around 0.2 at each maximum wavelength at room temperature by using a micro pipet. After titration, the combined solution was gently shaken several times to stabilize the absorbance and fluorescence intensities of all samples. (see detailed results in the supporting information)

**Fluorescence quantum yield.** The fluorescence quantum yields of dye-dsDNA complexes were determined with a Hamamatsu C9920-02 calibrated by the integrating sphere system. Measurements of absolute fluorescence quantum yields ($\Phi_F^{dsDNA}$) of PC dyes were performed with scan excitation mode at around maximum absorption wavelength ($\lambda_{abs}^{dsDNA}$), and values of $\Phi_F^{dsDNA}$ of monomethine dyes were calculated by using averaged values. Relative method was applied to determine the intrinsic fluorescence quantum yields ($\Phi_F^{free}$). The values of $\Phi_F^{free}$ for monomethine and trimethine PC are determined in relative to Rhodamine 6 G ethanol in solution ($\Phi_F = 0.94$)[42] and Cresyl violet in methanol

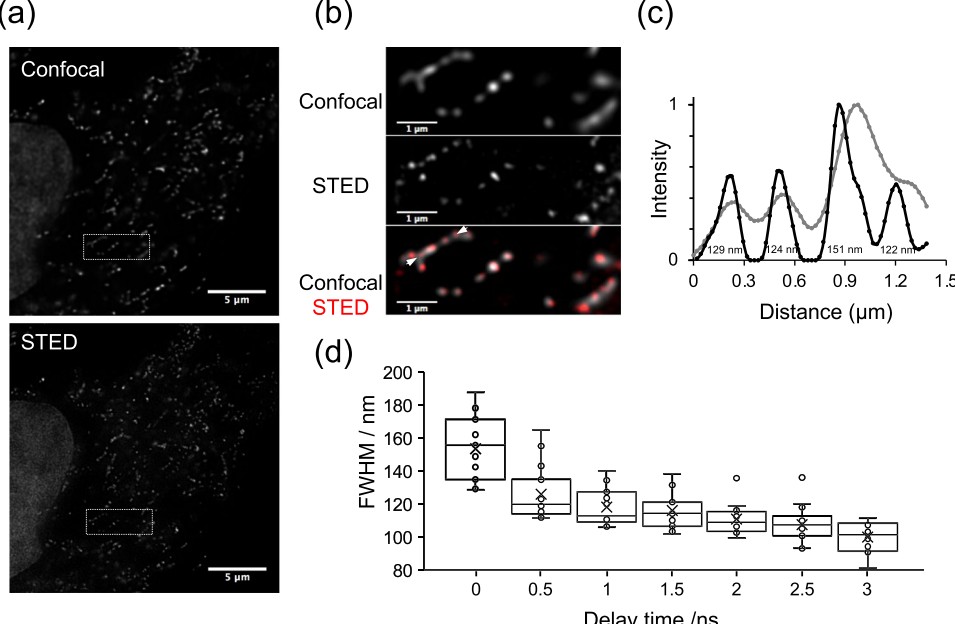

**Fig. 4 Comparison of confocal and SPLIT-STED imaging in living HeLa cells stained with PC3. a** Confocal and super-resolution images of mt-DNA stained with 10 nM **PC3** in living HeLa cells. Super-resolution images were obtained by two component separations using an n-exponential reconvolution model. **b** Enlarged images of the white dotted square region of **a**. The representative images are shown from five similar images. **c** An example of normalized fluorescence intensity profiles obtained from the region between arrows in **b**. Line profiles in STED and confocal image are shown in black and gray, respectively. FWHM values estimated by fitting with a Gaussian function are also indicated in the black line profile. **d** Box-whisker plots of FWHM value as a function of delay time in the minor axis of mt-nucleoids ($n = 15$ from three independent cells). Center line, median; box limits, upper and lower quartiles; whiskers, 1.5x interquartile range. Individual data points and the mean points are also indicated in white circle and black cross, respectively. Source Data is available as a source data file.

solution ($\Phi_F = 0.54$)[43] according to following equations.

$$\Phi_F^{\text{free}} = \Phi_F^{\text{reference}} \times (F^{\text{free-dye}}/F^{\text{reference}}) \times (A^{\text{reference}}/A^{\text{free-dye}}) \times (n^{\text{free-dye}}/n^{\text{reference}})^2$$

where $\Phi_F^{\text{referece}}$ represents fluorescence quantum yield of reference samples, $F^{\text{free-dye}}$ and $F^{\text{reference}}$ stand for the area of corrected fluorescence spectra of dyes and reference sample using same excitation wavelength, $A^{\text{reference}}$ and $A^{\text{free-Dye}}$ are the recorded absorbance of reference sample and dyes at the wavelength for the photoexcitation, and $n^{\text{free-dye}}$ and $n^{\text{reference}}$ represent the reflective index of the solvents for the measurements. Reflective index value of TE buffer solution was calculated using that of water ($n = 1.333$). The values of reflective index for ethanol ($n = 1.359$)[44] and methanol ($n = 1.327$)[44] were used for the detailed calculation.

**Fluorescence lifetime.** The light source used was a wavelength-tunable optical parametric amplifier based on a regeneratively amplified mode-locked Ti: sapphire laser with a pulse duration of 200 fs and a repetition rate of 200 kHz. The excitation wavelengths were set around the maximum absorption wavelengths for each sample. Emitted photons were detected using a single monochromator with an avalanche photodiode (SPD-050-CTE-N1; MPD). The detection wavelengths were tuned to the emission peak wavelengths for each sample. Each photon arrival times were recorded using time-correlated single-photon counting boards (SPC-130EM-N1; Becker & Hickl GmbH).

**Circular dichroism.** The measurements of CD were performed using FT/IR6100 (JASCO) with 1 cm square quartz cell. Before measurements, **PC1** and **PC5** were titrated with excess of dsDNA and are shaken several times.

**Animal and plant cell cultures for fluorescence imaging.** Cell culture lines, HeLa (RCB0007; RIKEN BRC), U-2OS (HTB96; ATCC), C6 (RCB2854; RIKEN BRC), and NIH3T3 (RCB1862; RIKEN BRC) were cultured in Dulbecco's modified Eagle's medium (DMEM, Wako) containing 10% fetal bovine serum at 37 °C in a 5% CO$_2$/95% air incubator. These lines ($2 \times 10^4$ cells/mL) were transferred on each well of a glass-bottom 8-well slide and cultured 1 day before imaging. DNA staining was performed in DMEM (−) containing 10 mM HEPES (pH 7.4) without washing. The *Arabidopsis thaliana* wild type (Col-0) was also used. After keeping at 4 °C for 3 days on Murashige and Skoog medium, seeds were cultured under continuous white light at 22–23 °C for germination and cultured for 7–12 days. For time-lapse imaging of root elongation, seedlings were grown between the cover glass and the medium[45]. Four milliliters of MS-agar medium was placed in a coverglass chamber (no. 5202001; IWAKI), the long side section (~5 × 45 mm) of

the agar was removed, and seeds were placed between the cover glass and the medium and incubated standing at an angle of about 70°.

**Time-lapse imaging for cell proliferation assay.** For assessment of cytotoxicity and phototoxicity of PC dyes, we also used commercialized red fluorescence nucleus markers (SYTO 80, SYTO 82, SYTO 84; ThermoFisher Scientific). HeLa cells were stained with each dye in DMEM (+) and the time-lapse observation was performed by an inverted microscope system (IX-71; Olympus) equipped with an UPlanSApo IR 20x/0.75 objective lens (Olympus), and a CMOS camera (ORCA Flash 4.0 V3 C13440; Hamamatsu photonics). The TRITC-A-Basic fluorescent filter set (FF01-542/20, FF570-Di01, FF01-620/52; Semrock Inc.) was used for all nucleus markers. The stage incubator system (Tokai Hit Co, Ltd.) was used to keep temperature at 37 °C and 5% CO$_2$/95% air condition. The fluorescence and bright-field time-lapse images were taken with or without excitation using an imaging software (MetaMorph; Molecular Devices) and cell proliferation rate was assessed by visual inspection from bright-field time-lapse images.

**Confocal microscopy.** A confocal laser scanning microscopy system (TCS SP8 FALCON gSTED; Leica) equipped with a pulsed white light laser (WLL; 80 MHz) and a HyD detector was used for fluorescence imaging of nuclear DNA in various animal cultured cells at 37 °C in a 5% CO$_2$/95% air condition (Fig. 2a, c). For low and high magnification observation, HC PL APO CS2 20×/0.75 and HC PL APO CS2 100×/1.40 oil objective were used, respectively. Cells stained with **PC1** or SYTO 80 were excited with a 532 nm and their emission was collected at 540–670 nm. Cells stained with **PC3** were excited with a 552 nm or a 561 nm and their emission was collected at 560–670 nm or 570–670 nm. When stained at ultralow concentration (100 pM), cells were excited with 561 nm and their emission was detected at 570–769 nm. Cells stained with SYTO 82 were excited with a 543 nm and their emission was detected at 550–670 nm. Cells stained with SYTO 84 were excited with a 561 nm and their emission was detected at 570–670 nm. Gated detection between 0.1–12 ns was performed for all fluorescence imaging. A confocal microscope system (LSM 7 Duo; Zeiss) equipped with a 20×/0.8 Plan-Apochromat lens and 32-channel gallium arsenide phosphide (GaASP) detector array was used for Arabidopsis leaf and root imaging. Cells stained with **PC1** were excited with a 514 nm and their fluorescence were collected at 517–614 nm in leaf cells and 517–693 nm in root cells, respectively. Cells stained with **PC3** were excited with a 560 nm and their fluorescence were collected at 561–605 nm in leaf cells and 570–693 nm in root cells, respectively. In leaf cell imaging, chlorophyll autofluorescence was also detected at 675–693 nm. Time-lapse imaging in Arabidopsis root was also performed every 5 min using a confocal laser scanning

microscope (CV1000; Yokogawa) equipped with a 20×/0.75. Collected images were further processed using open-source software ImageJ (http://imagej.nih.gov/ij/) and Adobe Photoshop CS6.

**Two-photon excitation microscopy**. Two-photon imaging was performed using a laser scanning microscope (LSM 7 Duo; Zeiss) equipped with a widely tunable Ti: Sapphire femtosecond pulse laser (Chameleon; Coherent) and LD C-Apochromat 40×/1.1 water immersion lens. The same Arabidopsis root stained with **PC1** were excited with 1000 nm as well as 488 nm and their fluorescence were detected at 500–690 nm and 490–596 nm, respectively.

**FLIM and SPLIT-STED microscopy**. A confocal laser scanning microscopy system (TCS SP8 FALCON gSTED; Leica) equipped with a pulsed white light laser (WLL; 80 MHz), 660 STED laser (Fig. 4 and Supplementary Figs. 12, 13), 775 STED laser (Supplementary Fig. 11) HC PL APO CS2 100×/1.40 oil objective lens, and a HyD detector was used for fluorescence imaging of nuclear and mt-DNA in various animal cultured cells at 37 °C in a 5% CO2/95% air condition (Fig. 4a–i). For observation of Arabidopsis stomata, HC PL APO CS2 93×/1.30 GLYC objective lens was used and z-sectioning image was obtained from 26 frames at 0.26 μm steps. For SPLIT-STED imaging, cells were excited with 561 nm and their emission was corrected at 570–650 nm with or without 660 nm STED laser. STED image was obtained by separation of a FLIM image to two exponential components thorough n-exponential reconvolution model or τ-STED function[22]. Confocal and STED imaging were acquired alternately between lines. Fluorescent lifetime based separation images were displayed with different pseudo colors by phaser plot analysis[22,46]. Collected images were deconvoluted by default setting of Huygens; signal-to-noise ratio and quality threshold were set to 7 and 0.05 for STED images, 20 and 0.05 for conventional CLSM images, respectively. Images were further processed using ImageJ. From the line profile obtained by ImageJ, FWHM was estimated by fitting with a Gaussian function[47].

**Co-staining of N-aryl PC dyes with Hoechst 33342 in living HeLa cells**. HeLa cells were co-stained with 100 nM PC dyes with various concentration of Hoechst 33342 (0, 100, 1, and 3 μM). The confocal images were obtained with a TCS SP8 (Leica) equipped with HC PL APO CS2 100×/1.40 oil objective. The image channels used were Ex 405 nm and Em 420–480 nm for Hoechst 33342, Ex 532 nm/Em 540–670 nm for **PC1**, and Ex 561 nm/Em 570–670 nm for **PC3**, respectively.

**Co-staining of PC dyes with a mitochondrial staining dye**. HeLa cells were co-stained with 1 nM PC dyes with 20 nM MitoTracker Deep Red. The confocal images were obtained with a TCS SP8 (Leica) equipped with HC PL APO CS2 100×/1.40 oil objective. The image channels used were Ex 514 nm and Em 520–610 nm for **PC1**, Ex 561 nm/Em 570–610 nm for **PC3**, and Ex 633 nm/Em 640–680 nm for MitoTracker Deep Red, respectively.

**Co-staining of PC dyes with a mt-DNA staining dye**. HeLa cells were co-stained with 1 nM PC dyes with 300,000-fold dilution of SG. The confocal images were obtained with a LSM780 equipped with Plan-Apochromat 63×/1.40 oil objective. The image channels used were Ex 514 nm and Em 517–695 nm. Linear-deconvolution images were obtained using each reference spectra of **PC1**, **PC3**, and SG.

**Time-lapse analysis of Arabidopsis root and root hairs**. Arabidopsis root stained with 1 μM **PC1** was observed every 5 min excited with 488 nm and the emission spectrum was collected through band-pass filter BP525/50. The fluorescence images are maximum z-projections of 20 planes (4.3 μm intervals).

**Chemical synthesis**. Unless otherwise noted, all materials including dry solvents were obtained from commercial suppliers and used without further purification. The starting materials such as 1-(2,4-dinitrophenyl)−4-methylpyridin-1-ium chloride[48] and **5a**[49] were prepared according to the reported procedures. All work-up and purification procedures were carried out with reagent-grade solvents. High-resolution mass spectra analyses were carried out using FT-ESI mass analyzer. Melting points of all compounds were measured on a MPA100 Optimelt automated melting point system. Column chromatography was performed with silica gel 60 N (Kanto Chemical Co., spherical, neutral, 40–50 mesh) for the purification of **PC1**, **PC2**, **PC3**, **PC4**, **PC5**, **PC6**, **PC7**, and **PC8**. Column chromatography was performed with amino silica gel (Fuji Sylysia Chemical LTD, Cat. No. Hu 41003) for the purification of **8a**, **8b**, **8c**, **8d**, **PC8**, and **PC9**. Purifications of the other compounds were performed by silica gel 60 N (Kanto Chemical Co., spherical, neutral, 40–100 mesh). The detail synthetic procedure of each N-aryl PC dye and nuclear magnetic resonance (NMR) spectra were available in the supporting information file. NMR spectra were recorded on JEOL JNM-ECA-400 (1H 400 MHz, 13C 100 MHz) and JEOL ECA 600II with Ultra COOL probe (1H 600 MHz, 13C 150 MHz) spectrometers in dimethyl sulfoxide-d6 (d-DMSO) ((CD3)2SO). Chemical shifts for 1H NMR are expressed in parts per million (ppm) relative to (CD3)2SO (δ 2.49 ppm). Chemical shifts for 13C NMR are expressed in ppm

relative to (CD3)2SO (δ 39.5 ppm). Data were reported as follows: chemical shift, multiplicity (s singlet, d doublet, dd doublet of doublets, ddd doublet of doublet of doublets, dt doublet of triplets, td triplet of doublets, t triplet, q quartet, quin quintet, m multiplet, br broad), coupling constant (Hz), and integration. The purity of all PC dyes used for DNA stains are demonstrated by 1H-NMR.

**Reporting Summary**. Further information on research design is available in the Nature Research Reporting Summary linked to this article.

## Data availability
The authors declare that all data of this study are available within the manuscript and its supplementary files are available from the corresponding author upon request. Source data are provided with this paper.

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

# ARTICLE

21. Singer, V. L., Jones, L. J., Yue, S. T. & Haugland, R. P. Characterization of PicoGreen reagent and development of a fluorescence-based solution assay for double-stranded DNA quantitation. *Anal. Biochem.* **249**, 228–238 (1997).
22. Lanzanò, L. et al. Encoding and decoding spatio-temporal information for super-resolution microscopy. *Nat. Commun.* **6**, 6701 (2015).
23. Leubner, I. H. Synthesis and properties of pyrido- and azapyridocyanines. *J. Org. Chem.* **38**, 1098–1102 (1973).
24. Tolbert, L. M. & Zhao, X. Beyond the cyanine limit: peierls distortion and symmetry collapse in a polymethine dye. *J. Am. Chem. Soc.* **119**, 3253–3258 (1997).
25. Latt, S. A. & Stetten, G. Spectral studies on 33258 Hoechst and related bisbenzimidazole dyes useful for fluorescent detection of deoxyribonucleic acid synthesis. *J. Histochem. Cytochem.* **24**, 24–33 (1976).
26. Breusegem, S. Y., Clegg, R. M. & Loontiens, F. G. Base-sequence specificity of Hoechst 33258 and DAPI binding to five (A/T)4 DNA sites with kinetic evidence for more than one high-affinity Hoechst 33258-AATT complex. *J. Mol. Biol.* **315**, 1049–1061 (2002).
27. Moon, J. H., Kim, S. K., Sehlstedt, U., Rodger, A. & Nordén, B. DNA structural features responsible for sequence-dependent binding geometries of Hoechst 33258. *Biopolymers* **38**, 593–606 (1996).
28. Kapuściński, J. & Szer, W. Interactions of 4′, 6-diamidine-2-phenylindole with synthetic polynucleotides. *Nucleic Acids Res.* **5**, 3775–3799 (1978).
29. Reus, C., Stolar, M., Vanderkley, J., Nebauer, J. & Baumgartner, T. A convenient N-arylation route for electron-deficient pyridines: the case of π-extended electrochromic phosphaviologens. *J. Am. Chem. Soc.* **137**, 11710–11717 (2015).
30. Uno, K. et al. Key structural elements of unsymmetrical cyanine dyes for highly sensitive fluorescence turn-on DNA probe. *Chem. Asian J.* **12**, 233–238 (2017).
31. Albota, M. et al. Design of organic molecules with large two-photon absorption cross sections. *Science* **281**, 1653–1656 (1998).
32. Williamson, D. H. & Fennell, D. J. Visualization of yeast mitochondrial DNA with the fluorescent stain 'DAPI'. *Meth. Enzymol.* **56**, 728–733 (1979).
33. Kristoffersen, A. S., Erga, S. R., Hamre, B. & Frette, Ø. Testing fluorescence lifetime standards using two-photon excitation and time-domain instrumentation: rhodamine B, coumarin 6 and lucifer yellow. *J. Fluoresc.* **24**, 1015–1024 (2014).
34. Bunting, J. R., Phan, T. V., Kamali, E. & Dowben, R. M. Fluorescent cationic probes of mitochondria. Metrics and mechanism of interaction. *Biophys. J.* **56**, 979–993 (1989).
35. Jakobs, S. & Wurm, C. A. Super-resolution microscopy of mitochondria. *Curr. Opin. Chem. Biol.* **20**, 9–15 (2014).
36. Ježek, P., Špaček, T., Tauber, J. & Pavluch, V. Mitochondrial nucleoids: superresolution microscopy analysis. *Int. J. Biochem. Cell Biol.* **106**, 21–25 (2019).
37. Kukat, C. et al. Super-resolution microscopy reveals that mammalian mitochondrial nucleoids have a uniform size and frequently contain a single copy of mtDNA. *Proc. Natl Acad. Sci. USA* **108**, 13534–13539 (2011).
38. Brown, T. A. et al. Superresolution fluorescence imaging of mitochondrial nucleoids reveals their spatial range, limits, and membrane interaction. *Mol. Cell Biol.* **31**, 4994–5010 (2011).
39. Stephan, T., Roesch, A., Riedel, D. & Jakobs, S. Live-cell STED nanoscopy of mitochondrial cristae. *Sci. Rep.* **9**, 12419 (2019).
40. Zipper, H., Brunner, H., Bernhagen, J. & Vitzthum, F. Investigations on DNA intercalation and surface binding by SYBR Green I, its structure determination and methodological implications. *Nucleic Acids Res.* **32**, e103 (2004).
41. Li, Q. et al. RNA-selective, live cell imaging probes for studying nuclear structure and function. *Chem. Biol.* **13**, 615–623 (2006).
42. Fischer, M. & Georges, J. Fluorescence quantum yield of rhodamine 6G in ethanol as a function of concentration using thermal lens spectrometry. *Chem. Phys. Lett.* **260**, 115–118 (1996).
43. Magde, D., Brannon, J. H., Cremers, T. L. & Olmsted, J. III Absolute luminescence yield of cresyl violet. A standard for the red. *J. Phys. Chem.* **83**, 696–699 (1979).
44. Albuquerque, L., Ventura, C. & Gonçalves, R. Refractive indices, densities, and excess properties for binary mixtures containing methanol, ethanol, 1,2-ethanediol, and 2-methoxyethanol. *J. Chem. Eng. Data* **41**, 685–688 (1996).
45. Nakamura M. et al. Auxin and ROP GTPase signaling of polar nuclear migration in root epidermal hair cells. *Plant Physiol.* **176**, 378–391 (2018).
46. Digman, M. A., Caiolfa, V. R., Zamai, M. & Gratton, E. The phasor approach to fluorescence lifetime imaging analysis. *Biophys. J.* **94**, L14–L16 (2008).
47. Wang, C. et al. A photostable fluorescent marker for the superresolution live imaging of the dynamic structure of the mitochondrial cristae. *Proc. Natl Acad. Sci. USA* **116**, 15817–15822 (2019).
48. Genisson, Y., Marazano, C., Mehmandoust, M., Gnecco, D. & Das, B. C. Zincke's reaction with chiral primary amines: a practical entry to pyridinium salts of interest in asymmetric synthesis. *Synlett* 431–434 (1992).
49. Coe, B. J. et al. Quadratic nonlinear optical properties of N-aryl stilbazolium dyes. *Adv. Funct. Mater.* **12**, 110–116 (2002).

## Acknowledgements

We are deeply grateful to Prof. K. Itami and Prof. T. Higashiyama at Nagoya University for enormous help of this work such as invaluable comments, discussion, and financial support. We thank Dr. H. Ito, Dr. Y. Segawa, Dr. T. Fujikawa, and Dr. K. Kato at Nagoya University for many helpful comments. We also thank M. Tsuzuki (Nagoya University) for supporting cell culture and cell toxicity assessment. This work was supported by Grant-in-Aid for Scientific Research from JSPS (14J03652 to K.U., 19H05364, 20H05412 to Y.S., 16H06464, 16K21727, JP16H06280 to T.H.), Toyoaki scholarship Foundation and Ohsumi Frontier Science Foundation to Y.S., the ERATO program from JST (JPMJER1302 to K.I.) and the CREST program from JST (JPMJCR1924 to Y.Tsuchiya, Nagoya University).

## Author contributions

K.U. and Y.S conceived and designed this research. K.U performed synthesis of all PC dyes and most of the spectroscopic measurements. Y.S. and N.S. performed fluorescence titration experiments using hairpin oligonucleotides and all imaging experiments. K.U. and Y.S. wrote the manuscript. All authors read and approved the manuscript.

## Competing interests

The patent application, "Cyanine compound and fluorescent dye" (WO2019012963) invented by Y.S. and K.U. has been published (https://patentscope2.wipo.int/search/en/detail.jsf?docId=WO2019012963). N.S. declares no competing interests.
