## [Peer Review File · Nature Communications]

Reviewers' Comments:

Reviewer #1:

Remarks to the Author:

The authors present the pyrido cyanine molecules as molecular probes for live cell imaging of DNA. The idea of pyrido cyanine was in principle first followed by Gutsulyak and Romanko, when they synthesized methine dyes from 1-aryl-3-acetylpyridinium salts and by Leubner with 2,2'-, 2,4'-, and 4,4'-pyrido- and -azapyridocyanines. In this manuscript additional pyrido cyanines were synthesized and tested as DNA markers. The pyridine cyanines with the N-Aryl residues bind with high selectivity to DNA. The advantage of the developed dyes compared to the established Hoechst 33342 are that they allow to discriminate between mt-DNA and nuclear DNA. Although the manuscript provides results that are of interest, the paper would be more appropriate for a specialized journal in the field.

Reviewer #2:

Remarks to the Author:

In the manuscript, the authors developed a new class of fluorescent probe for DNA compatible to two-photon, life-time and STED mode of super-resolution imaging. The probes are composed of N-aryl pyrido cyanine scaffold with a long absorption wavelength. The high DNA specificity relative to RNA and the less toxic property, good membrane permeability are beneficial for practical application. Indeed, the authors showed a few of examples imaging DNA in various cell lines including plant tissues. However, this referee is frustrated with the heavy lack of scientific discussion in this paper. For example, the authors claimed the high selectivity of probe to DNA over RNA in term of the fluorogenic property, but the reason is not discussed at all. Similarly, its DNA binding mode and AT base-pair preference are not clearly explained. Also, the author used the lifetime difference of PC1 between the nucleus and mitochondria for the selective imaging, however the mechanism of such difference is not described.

Overall, indeed technically sound in many aspects, this referee judges that the content of this paper doesn't reach to the standard level of Nature Communications. I recommend it to be submitted to a more specialized journal.

Reviewer #3:

Remarks to the Author:

The report by Uno et al presents the properties of a class of DNA fluorescent markers, briefly called PC dyes.

The results show that these dyes have excellent properties in terms of DNA staining selectivity, spectral properties, limited cytotoxicity, compatibility with advanced microscopy techniques.

The authors show significant improvements compared to the state-of-the art DNA binding dye, SiR-Hoechst:

DNA binding properties unaltered, no requirement of verapamil to get uniform staining of the nucleus (this limitation of SiR-Hoechst should also be stated in the introduction)

In light of this, I think that the paper is of high interest for the community of fluorescence microscopists, and thus for the broad audience of Nature Communications.

Technical comments:

The characterization of the PC dyes is thorough and systematic. The results provided by the different microscopy techniques are convincing. However, the introduction can be a little expanded to describe in a more comprehensive way the state-of-the-art. For instance, surprisingly I don't see any mention to the Rhodamine-Hoechst positional isomers (Bucevičius et al, Chemical Science 2019).

I think the term "STED-FLIM" is misleading and should not be used by the authors.

The authors perform STED imaging with a time-resolved commercial instrument, the Leica sp8 Falcon. The lifetime information is used ONLY to improve the resolution of the STED image. This approach was first introduced in a paper in Nature Communications (Lanzanò et al, Nat Commun 2015, <https://www.nature.com/articles/ncomms7701>) where it was called SPLIT (separation of photons by lifetime tuning) and today is implemented in the Leica sp8 Falcon.

The authors should use a different term, for instance, "time-resolved STED" (in opposition to time-gated STED) or the more specific "SPLIT-STED" or "phasor-STED" or even the commercial name "tau-STED" and give credit to the abovementioned paper.

Feb 25th 2021

Response to Reviewers

We appreciate helpful comments from reviewers. We have carefully revised the manuscript according to the comments. Modified parts are shown with red characters in the text.

Reviewer #1:

The authors present the pyrido cyanine molecules as molecular probes for live cell imaging of DNA. The idea of pyrido cyanine was in principle first followed by Gutsulyak and Romanko, when they synthesized methine dyes from 1-aryl-3-acetylpyridinium salts and by Leubner with 2,2', 2,4', and 4,4'-pyrido- and -azapyridocyanines. In this manuscript additional pyrido cyanines were synthesized and tested as DNA markers. The pyridine cyanines with the N-Aryl residues bind with high selectivity to DNA. The advantage of the developed dyes compared to the established Hoechst 33342 are that they allow to discriminate between mt-DNA and nuclear DNA. Although the manuscript provides results that are of interest, the paper would be more appropriate for a specialized journal in the field.

Thank you for your comments, but we can't help but think that the assessment of the novelty is not properly conducted. As this reviewer pointed out, there have been previous reports on the synthesis of pyrido cyanine skeleton derivatives and we cite an appropriate paper in the text (Leubner, 1973). The novelty to be assessed in our manuscript is not the chemical structure itself, but rather its overwhelmingly superior properties as DNA probes (DNA selectivity, long-wavelength, cell permeability, organellar DNA stainability, and compatibility with advanced microscopes) when compared to widely known conventional DNA staining dyes, such as Hoechst and SYBR-Green. Given the importance of DNA fluorescent dyes as a research tool for many researchers in the life sciences, we believe our work makes a big leap in the field of molecular and cellular biology. Therefore, we are no doubt that the novelty of our work is suited to *Nature Communication*.

Reviewer #2:

the authors claimed the high selectivity of probe to DNA over RNA in term of the fluorogenic property, but the reason is not discussed at all. Similarly, its DNA binding mode and AT base-pair preference are not clearly explained.

Thank you for these important comments. In response to these comments, first, we examined a competitive experiment in DNA sequence using ^{AATT}DNA hairpin oligo and found that **PC1** shares the same ^{AATT}DNA sequences with Hoechst (**Fig. S3k**). We also examined circular dichroism spectrum of **PC1** complexed with dsDNA. As a result, The CD spectrum clearly showed a positive cotton effect in both dyes (**Fig. S4**), similar to that of minor-groove binders, such as Hoechst and DAPI, suggesting that **PC1** binds to dsDNA with minor-groove. The similar binding mode to Hoechst would contribute the high selectivity of **PC1** to DNA over RNA. Furthermore, to assess the reason why the DNA selectivity is higher than that of Hoechst, effects of (methyl)piperazine site, which is part of Hoechst, on fluorogenic properties was investigated. We synthesized a PC dye (**PC 9**) with a (methyl)piperazine site by replacing one of the two diethylamino groups of **PC4** with (methyl)piperazine and examined the fluorogenic properties (**Fig. S2i**). As a result, the fluorescent intensity of **PC 9** increased 250-fold upon binding RNA, which was 5 times higher than that of **PC4** (53-fold) (**Fig. S2d, i**), suggesting that the binding of the (methyl) piperazine moiety to RNA is thought to be a factor that reduces DNA selectivity. We have mentioned these new results in the text in revised manuscript.

the author used the lifetime difference of PC1 between the nucleus and mitochondria for the selective imaging, however the mechanism of such difference is not described.

Thank you for this careful comment. The fluorescent lifetime depends on its energy dispersal to the surrounding environment. There are many possibilities, especially in living cells, but one major factor is the shortening of the fluorescent lifetime due to self-absorption between fluorescent dyes in the case of dye properties with small stokes-shift (Kristoffersen et al. 2014). First, to investigate whether the fluorescent lifetime of **PC1** changes at high concentrations, we measured the fluorescent lifetime of **PC1** at various concentrations of **PC1** using a constant concentration of pBluescript II SK+ DNA vector. As a result, the fluorescent lifetime of **PC1** was found to be shortened in a concentration-dependent manner possibly because of self-absorption (**Fig. S9a**). On the other hand, it is known that cationic fluorescent dyes tend to accumulate in mitochondria, which have membrane potential (Bunting et al. 1989 Biophys J). In fact, **PC1** stains mitochondria DNA at less than 100 pM, while more than 1 nM of

PC1 is required for staining nuclear DNA (**Fig. 3**). This suggests that the dye concentration is different between nuclear and mitochondrial compartments, even when cells were stained at a constant concentration. Therefore, it is possible that the increase in dye concentration may cause a shortening of the fluorescent lifetime in mitochondria due to self-absorption more easily than in the nucleus. To test this possibility, we stained cells with different **PC1** concentrations and measured the fluorescent lifetime of **PC1** in nuclear DNA and mitochondria DNA. We found that the fluorescent lifetime of **PC1** in nuclear DNA was maintained up to 10 nM, while that in mitochondrial DNA was shortened above 100 pM (**Fig. S9b**). Taken together with these results, it is considered that the fluorescent lifetime of **PC1** between nuclear DNA and mitochondrial DNA can be separated when staining at 1 nM, because the fluorescent lifetime in mitochondrial DNA is shortened due to self-absorption of **PC1**, while it is retained in nuclear DNA. We have mentioned these new results in the text in revised manuscript.

References

1. Kristoffersen, A. S., Erga, S. R., Hamre, B. & Frette Ø. Testing Fluorescence Lifetime Standards using Two-Photon Excitation and Time-Domain Instrumentation: Rhodamine B, Coumarin 6 and Lucifer Yellow. *J. Fluoresc.* **24**, 1015-1024 (2014).
2. Bunting, J. R., Phan, T. V., Kamali, E., Dowben, R. M. Fluorescent cationic probes of mitochondria. Metrics and mechanism of interaction. *Biophys. J.* **56**, 979–993 (1989).

Reviewer #3:

The introduction can be a little expanded to describe in a more comprehensive way the state-of-the-art. For instance, surprisingly I don't see any mention to the Rhodamine–Hoechst positional isomers (Bucevičius et al, Chemical Science 2019).

Thank you for this kind and helpful comment. We agree with this comment and have cited the paper as the reference #14 in line 14 and 16 of page 2.

I think the term "STED-FLIM" is misleading and should not be used by the authors.

Thank you for this kind and valuable comment. We agree this comment and have corrected the term "STED-FLIM" to "SPLIT-STED", citing the paper you mentioned.

Yours sincerely,

Yoshikatsu Sato

Institute of Transformative Bio-Molecules (WPI-ITbM)

Graduate School of Science, Nagoya University

E-mail: sato.yoshikatsu@i.mbox.nagoya-u.ac.jp

Phone/Fax: +81-52-789-2970

Reviewers' Comments:

Reviewer #3:

Remarks to the Author:

The authors have replied to my technical comments and accepted my suggestions.

They have also replied to reviewer 1 regarding the novelty of the paper: I agree with the authors that the importance of the paper is not on the chemistry itself but on the superior properties of the reported DNA probes, and thus their huge potential in the applications. I confirm that, in my opinion, the paper is of high interest for the community of fluorescence microscopists/biologists, and thus for the audience of Nature Communications.

In the revised version of the paper, Uno et al have included additional data in reply to the comments of reviewer 2. These data further strengthen the paper by suggesting mechanisms that affect the lifetime of the probes (e.g. concentration), their fluorogenic properties and their DNA binding properties.

In light of all that, I definitely recommend acceptance of the paper on Nature Communications, as is.

Response to Reviewers

Reviewer #3:

The main text has also been modified to include the discussion of the points raised by the reviewer 2. The added text is clear, except for a language-typo at page 5 line 30 (" ... because PC dyes with short stokes-shift are cationic, which is easily accumulate in mitochondria (Fig. S9) ")

We appreciate the kind comments on the points raised by reviewer 2. We have changed "because PC dyes with short stokes-shift are cationic, which is easily accumulate in mitochondria" to "because PC dyes with short stokes-shifts are cationic and tend to accumulate in mitochondria".

Yours sincerely,

Yoshikatsu Sato

Institute of Transformative Bio-Molecules (WPI-ITbM)

Graduate School of Science, Nagoya University

E-mail: sato.yoshikatsu@i.mbox.nagoya-u.ac.jp

Phone/Fax: +81-52-789-2970